



# Extending the applicability of a wind-farm gravity-wave model to vertically non-uniform atmospheres

Koen Devesse[1], Luca Lanzilao[1], Sebastiaan Jamaer[2], Nicole van Lipzig[2], and Johan Meyers[1]

[1]Department of Mechanical Engineering, KU Leuven, Leuven, Belgium
[2]Department of Earth and Environmental Sciences, KU Leuven, Leuven, Belgium

**Correspondence:** Koen Devesse (koen.devesse@kuleuven.be)

**Abstract.** Recent research suggests that atmospheric gravity waves can affect off-shore wind farm performance. A fast wind-farm boundary-layer model has been proposed to simulate the effects of these gravity waves on wind-farm operation by Allaerts and Meyers (2019). The current work extends the applicability of that model to free atmospheres in which wind and stability vary with altitude. We validate the model using reference cases from literature on mountain waves. Analysis of two reference flows shows that internal gravity wave resonance caused by the atmospheric non-uniformity can prohibit perturbations in the ABL at the wavelengths where it occurs. To determine the overall impact of the vertical variations in the atmospheric conditions on wind farm operation, we consider one year of operation of the Belgian–Dutch wind-farm cluster with the extended model. We find that this impact on individual flow cases is often of the same order of magnitude as the total flow perturbation. In 16.5% of the analysed flows, the relative difference in upstream velocity reduction between uniform and non-uniform free atmospheres is more than 30%. However, this impact is small when averaged over all cases. This suggests that variations in the atmospheric conditions should be taken into account when simulating wind-farm operation in specific atmospheric conditions.

## 1 Introduction

In recent years, it has been well documented that wind farms form a blockage to the flow in and around them (Bleeg et al., 2018), thereby displacing the Atmospheric Boundary Layer (ABL). Such displacements can propagate through the overlying inversion layer and free atmosphere as waves in stably stratified atmospheres, conditions which frequently occur at sea. As offshore wind farms in Europe increase in size and installed capacity (WindEurope, 2018), improving the understanding and simulation of gravity-wave wind-farm interaction becomes crucial to optimizing turbine control and wind farm layout (see, eg. Lanzilao and Meyers (2021b)).

Previous work on the interaction between gravity waves and wind farms has assumed the free atmosphere to be uniformly stratified, with a constant background wind (Smith, 2010; Allaerts and Meyers, 2017, 2019). However, for waves with horizontal scale of a few kilometers, such as those triggered by wind farm blockage, vertical variations of the atmospheric conditions





can drastically influence the pressure feedback they induce (Teixeira et al., 2013). Therefore, the goal of this work is to determine how variations of the free atmosphere's properties with altitude affect the interaction between gravity waves and the flow

in the ABL, and overall wind farm performance.

Allaerts and Meyers (2019) proposed a model focused on the interaction between wind turbines and gravity waves. Based on earlier work by Smith (2010), the model explicitly simulates the flow in the ABL as two height-averaged horizontal layers, and incorporates the free atmosphere as a boundary condition. This boundary condition links variations in ABL height to the pressure gradients induced by the corresponding gravity waves. In Fourier space, the relation between these height and

pressure perturbations is defined by the stratification coefficient $\hat{\Phi}$. As the model is based on a three-layer representation of the atmosphere, it is called the Three-Layer Model (TLM).

Currently, the TLM can only describe uniformly stratified free atmospheres, which places a strong restriction on the atmospheric conditions that can be represented. This work adapts the TLM for flow profiles that vary with altitude, and studies how these variations change the interaction between the ABL flow and gravity waves. A common approach in gravity wave theory

is to use a piecewise representation of the upper atmosphere, where the profiles of the stratification and the wind speed are split up in a discrete number of layers (Tolstoy, 1973; Gossard and Hooke, 1975; Baines, 1998; Smith et al., 2002; Teixeira et al., 2013; Yu and Teixeira, 2015). We generalize this approach by allowing for an arbitrarily large amount of layers, so that any atmospheric profile can be accurately analyzed.

It is well known that variations in the atmospheric state can cause wave reflection, which might lead to internal gravity-

wave resonance (Gill, 1982). Using the extended TLM, we examine how this resonance affects the pressure feedback on the ABL. Extensive research has been performed on internal gravity-wave resonance, with most of it focusing on flow around topographies, mountain waves, and the effect on the free atmosphere (Teixeira, 2014). As a result, the vertical displacement is simply given by the shape of the topography, not by a change in ABL height. In wind farms, this is different: the displacement is given by a change in ABL depth due to wind farm blockage, in which the induced gravity waves can play an active role. We

therefore set up a qualitative analysis of the feedback loop between ABL flow perturbations and the associated resonant gravity wave feedback. Finally, in order to estimate the overall effect of vertical variations in the atmospheric conditions, one year of operation of the Belgian-Dutch wind-farm cluster is simulated.

The remainder of this paper is organized as follows. Section 2 reviews how gravity waves and the ABL flow interact, and how the TLM models ABL flow. In Sect. 3, the analytical expressions for the stratification coefficients for non-uniform atmospheres

are derived. To evaluate these, a method for solving the wave equation has to be developed, which is done in Sect. 3.2. Section 4 uses the developed method to analyze the effect of variations of the wind and stability in the free atmosphere on wind-farm operation. Finally, the paper concludes with a summary and suggestions for further research in Sect. 5.

## 2   Gravity wave interaction with wind farms

Wind farms form a blockage to the ABL flowing through and around them, thereby pushing the inversion layer, and the free

atmosphere above, upwards. These displacements can trigger gravity waves, which may influence the ABL flow by inducing



pressure gradients. The first part of this section discusses these induced pressures, while the second part gives an overview of how the TLM models ABL flow, and how it incorporates gravity wave effects..

## 2.1 Pressure feedback induced by gravity waves

The vertical displacement $\eta_t$ of the capping inversion topping the ABL leads to two types of atmospheric gravity waves. The first type, called *inversion gravity waves*, propagate horizontally along the inversion layer, similar to surface water waves. Due to the strong stratification in the inversion layer, any vertical displacement $\eta_t$ of the inversion layer will be counteracted by the buoyancy-generated changes in pressure $p_1'$ (Gill, 1982; Smith, 2010):

$$\frac{p_1'}{\bar{\rho}} = g'\eta_t \tag{1}$$

where $\bar{\rho}$ is the unperturbed density and $g' = g\Delta\theta/\theta$ is the reduced gravity, determined by the potential temperature $\theta$ and the inversion strength $\Delta\theta$.

The second type of waves propagates vertically through the free atmosphere above the capping inversion if it is stably stratified, as is usually the case. The free atmosphere perceives $\eta_t$ similarly to large-scale topographies, and *internal gravity waves* are generated (Smith, 2010). In return, the internal gravity waves also induce pressure gradients on the ABL. This pressure feedback is easily expressed by using double Fourier transforms in the horizontal directions (i.e. $(x, y) \rightarrow (k, l)$) of the displacement and the pressure. For each wavenumber $(k, l)$, the perturbation in the free atmosphere is a plane wave, with the vertical velocity given by (Nappo, 2012):

$$w'(x, y, z) = -\imath\Omega\hat{\eta}_t \exp\left[\imath(kx + ly + mz)\right] \tag{2}$$

where $\Omega = -\boldsymbol{\kappa} \cdot \bar{\boldsymbol{u}} = -(\bar{u}k + \bar{v}l)$ is the intrinsic frequency of the gravity waves, with $\bar{u}$ and $\bar{v}$ the $x$ and $y$ components of the background velocity, and where $m$ is the vertical wavenumber of the internal gravity waves. For each separate wavenumber $(k, l)$, the pressure perturbation $\hat{p}_2'$ of the plane wave is proportional to the ABL displacement $\hat{\eta}_t$, with the relation determined by the stratification coefficients $\hat{\Phi}$ (Smith, 2010):

$$\frac{\hat{p}_2'}{\bar{\rho}} = \hat{\Phi}\hat{\eta}_t. \tag{3}$$

For uniform free atmospheres, these coefficients are given by (Smith, 2010):

$$\hat{\Phi} = \frac{\imath\left(N_g^2 - \Omega^2\right)}{m} \tag{4}$$

where $N_g = \sqrt{\frac{g}{\theta}\frac{d\theta}{dz}}$ is the Brunt-Väisälä frequency, which is constant with height in the uniform case. Further, $m$ can be found using the dispersion relation (Gill, 1982):

$$m^2 = \left(k^2 + l^2\right)\left(\frac{N_g^2}{\Omega^2} - 1\right) \tag{5}$$

If $\Omega^2 < N_g^2$, then $m^2$ is positive, and the waves propagate vertically. If $\Omega^2 > N_g^2$, then $m^2$ is negative, and the waves become evanescent. To find $\Phi$, the sign of $m$ has to be known. For propagating waves ($m^2 > 0$), the sign of $m$ is determined by the





*radiation condition*, which states that the energy flux of a wave, which is directed along $\frac{\partial \Omega}{\partial m}$, should be upwards, away from the perturbation (Baines, 1998). It then follows that: $\text{sign}(m) = -\text{sign}(\Omega)$. For evanescent waves ($m^2 < 0$), the positive root has to be chosen for the perturbation to die out with altitude (Smith, 1980).

## 2.2 Three-layer model

The TLM is based on earlier work by Smith (2010), who first analyzed the impact of gravity waves on wind-farm operation.
It represents the ABL as being neutrally stable and capped by an inversion layer, with the flow above being stably stratified. Although stably stratified atmospheres have often been modeled this way (Klemp and Lilly, 1975; Durran, 1990; Vosper, 2004; Smith, 2007; Sachsperger et al., 2015), Smith was the first to apply such a model to wind-farm operation. His model assumes that the flow in the ABL does not vary with height, so that the perturbed variables can be replaced by their height averages. As a result, horizontal flow divergence and convergence will lead to changes in the ABL height. Another assumption is that the
ABL flow is assumed to be hydrostatic. The inversion layer is modelled as a zero order jump in potential temperature $\Delta\theta$ at the top of the ABL, and the free atmosphere above is assumed to be uniformly stratified.

The TLM improves on the model by Smith in several ways, the most important of which is to divide the ABL in two separate layers. The resulting lower and an upper layer are denoted by subscripts 1 and 2, respectively. The two layers are separated by a pliant surface, similar to the interface between the ABL and the free atmosphere. The wind-farm forcing terms are added in the
momentum equations for the lower layer, while only affecting the upper layer through interaction through the pliant surface. For this reason, the lower layer is also called the wind-farm layer. The flow in the two layers is governed by the following two-dimensional depth-averaged linearised momentum and continuum equations (Allaerts and Meyers, 2019):

$$\bar{u}_{j,1}\frac{\partial u'_{i,1}}{\partial x_j} + \frac{1}{\bar{\rho}}\frac{\partial (g'+\Phi)*\eta_t}{\partial x_i} = f_c \epsilon_{ij3} u'_{j,1} + \nu_{t,1}\frac{\partial^2 u'_{i,1}}{\partial x_j \partial x_j} + \frac{D_{ij}}{H_1}\Delta_1^2 u'_j - \frac{C_{ij}}{H_1}u'_{j,1} + \frac{F_i}{H_1} \tag{6}$$

$$\bar{u}_{j,2}\frac{\partial u'_{i,2}}{\partial x_j} + \frac{1}{\bar{\rho}}\frac{\partial (g'+\Phi)*\eta_t}{\partial x_i} = f_c \epsilon_{ij3} u'_{j,2} + \nu_{t,2}\frac{\partial^2 u'_{i,2}}{\partial x_j \partial x_j} - \frac{D_{ij}}{H_1}\Delta_1^2 u'_j \tag{7}$$

$$\bar{u}_{j,1}\frac{\partial \eta_1}{\partial x_j} + H_1 \frac{\partial u'_{j,1}}{\partial x_j} = 0 \tag{8}$$

$$\bar{u}_{j,2}\frac{\partial \eta_2}{\partial x_j} + H_2 \frac{\partial u'_{j,2}}{\partial x_j} = 0 \tag{9}$$

The atmospheric base state is governed by the mean depth-averaged wind speeds $\bar{u}_{i,1}$ and $\bar{u}_{i,2}$ (with $i = 1, 2$) and the layer heights $H_1$ and $H_2$ of the wind-farm and upper layer, respectively, and $u'_{i,1}$, $u'_{i,2}$, $\eta_1$, and $\eta_2$ represent the perturbations to this reference state. The total inversion layer displacement $\eta_t$ is given by the sum of $\eta_1$ and $\eta_2$. Following Allaerts and
Meyers (2019), $H_1$ is taken to be twice the hub height of the turbines throughout this work. Further, $\Delta_1^2 u'_j = u'_{j,2} - u'_{j,1}$ is the perturbation of the velocity difference between the wind-farm and the upper layer, linked to the perturbation of the friction at the interface by the matrix $D_{ij}$. Similarly, the matrix $C_{ij}$ relates the perturbation of the friction at the ground to the velocity perturbations in the lower layer. Finally, $f_c$ is the Coriolis parameter, and $F_i$ is the wind turbine forcing term.

In equations 6–9, the pressure has been substituted using the equations discussed in Sect. 2.1. As the ABL is assumed to be
hydrostatic, the pressure perturbation is equal to the pressure induced by the gravity waves generated by the changes in ABL



height $\eta_t = \eta_1 + \eta_2$:

$$\frac{p'}{\rho} = (g' + \Phi) * \eta_t \tag{10}$$

where $*$ is the convolution operator. In this way, the TLM incorporates the gravity-wave effects through a pressure boundary condition, which is determined by the stratification coefficients $\Phi$. These coefficients do not depend on the values of $\eta_t$ or $p'$,
and can thus be calculated separately from the set of equations 6 to 9. To avoid the computationally expensive convolution, the TLM is solved using a spectral method with a Fourier–Galerkin discretisation (Allaerts and Meyers, 2019). Finally, we note that the hydrostatic assumption in the boundary layer ($\frac{\partial p}{\partial z} = 0$) is only reasonable as long as pressure effects of the gravity waves are significant. In particular in cases where a capping inversions is absent, we have noticed that this assumption is not valid, which can lead to perturbations that become unphysical. Therefore, in the current work, we will only consider atmospheric
cases with a capping inversion. Extension of the boundary layer equations (6, 7) to include for hydrodynamic effects is a topic of further research.

The turbines are represented individually using an actuator disk model. To incorporate their interactions, the TLM is coupled with a wake model, which in this work is a Gaussian wake model coupled with linear superposition of velocity deficits (Bastankhah and Porté-Agel, 2014; Niayifar and Porté-Agel, 2016).The forces $f_{i,k}$ for each individual turbine $k$ are written
as a first-order Taylor expansion around the background inflow velocity $\bar{u}_{fs}$, in order to incorporate the effect of the velocity perturbation (Allaerts and Meyers, 2019):

$$f_{i,k}(u_{fs}) = f_{i,k}(\bar{u}_{fs}) + J_i^{f,k}(u'_{fs}) \tag{11}$$

where $J_i^{f,k}$ is the Jacobian of $f_{i,k}$. The background inflow velocity is taken to be the mean wind speed in the wind farm layer, while the velocity perturbation is evaluated at a distance of $10D$ upstream of the farm, where $D$ is the diameter of the turbines.
Finally, the turbine forces are filtered on the numerical grid with a Gaussian filter (Allaerts and Meyers, 2019):

$$F_i = \int\limits_0^{L_x} \int\limits_0^{L_y} G(x - x', y - y') \sum_k^{N_t} f_{i,k} \delta(x - x_k, y - y_k) dx' dy' \tag{12}$$

where $L_x \times L_y$ is the size of the domain, $(x_k, y_k)$ denote the positions of the turbines, and $G(x - x', y - y')$ is a 2D Gaussian kernel (Allaerts and Meyers, 2019):

$$G(x, y) = \frac{1}{\pi L^2} \exp\left(-\frac{x^2 + y^2}{L^2}\right). \tag{13}$$

We use a filter length of $L = 1\text{km}$.

## 3 Extension to vertically non-uniform free atmospheres

In reality, the free atmosphere is not uniform, and the stratification strength and wind speed can strongly depend on altitude. This will of course impact internal gravity wave propagation through the atmosphere, and thus the pressure feedback of these





waves in the ABL. Currently, the TLM does not incorporate this, as the simplified version of the internal wave equation on

which equation 4 is based is only valid for uniformly stratified free atmospheres with a constant wind velocity. This section derives expressions for the stratification coefficients for vertically non-uniform atmospheres, where both the stratification strength and wind speed can vary.

### 3.1 Gravity waves in vertically non-uniform flows

The internal gravity wave equation in vertically non-uniform atmospheres with continuous background velocities is (Teixeira,

150 2014):

$$\left[\frac{D^2}{Dt^2}\left(\frac{\partial^2}{\partial x^2} + \frac{\partial^2}{\partial y^2} + \frac{\partial^2}{\partial z^2}\right) - \frac{D}{Dt}\frac{\partial^2}{\partial z^2}\left(\frac{D}{Dt}\right) + N_g^2\left(\frac{\partial^2}{\partial x^2} + \frac{\partial^2}{\partial y^2}\right)\right]w' = 0 \tag{14}$$

where $w'$ is the vertical velocity of the wave. For plane waves, the solution for this equation can be written as (Teixeira, 2014):

$$w'(x,y,z) = W(z)\exp\left[\imath(kx + ly)\right]. \tag{15}$$

Equation 14 then reduces to the Helmholtz equation (Gill, 1982):

$$\frac{d^2}{dz^2}W(z) + m^2 W(z) = 0 \tag{16}$$

where $m^2$ is given by:

$$m^2 = (k^2 + l^2)\left(\frac{N_g^2}{\Omega^2} - 1\right) - \Omega^{-1}\frac{d^2\Omega}{dz^2}. \tag{17}$$

It is important to note that $m^2$ is not a constant, as it depends on the altitude through $N_g(z)$ and $\Omega(z)$. The above derivation is only valid if the vertical background velocities are continuous. In points where the background velocities or their derivatives

to altitude have a discontinuity, $w'$ itself can be discontinuous, as will be elaborated in Sect. 3.2.2.

The relation between $w'$ and $p'$ is given by:

$$\left[\frac{D}{Dt}\frac{\partial}{\partial z} - \left(\frac{\partial\bar{u}}{\partial z}\frac{\partial}{\partial x} + \frac{\partial\bar{v}}{\partial z}\frac{\partial}{\partial y}\right)\right]w' = \left(\frac{\partial^2}{\partial x^2} + \frac{\partial^2}{\partial y^2}\right)\frac{p'}{\bar{\rho}} \tag{18}$$

For plane waves, this relation is:

$$\frac{\hat{p}'}{\bar{\rho}} = \frac{\imath}{k^2 + l^2}\left(\Omega\frac{dW}{dz} - \frac{d\Omega}{dz}W\right) \tag{19}$$

Using $W(H) = -\imath\Omega\hat{\eta}_t$ (cf. equation 2), the definition of the stratification coefficient then leads to:

$$\hat{\Phi} = \frac{\Omega}{k^2 + l^2}\left(\Omega\frac{dW}{dz} - \frac{d\Omega}{dz}W\right) \tag{20}$$

It is easily verified that the above equation and the expression for $m^2$ (equation 17) simplify to equations 4 and 5 if the free atmosphere is uniformly stratified.





## 3.2 Piecewise methods

To evaluate the expression for the stratification coefficients derived in the previous section, the Helmholtz equation for vertically non-uniform atmospheres has to be solved. This is no longer trivial, as the vertical wavenumber now varies with altitude. In earlier studies, one of the main approaches to solving the Helmholtz equation has been the so-called piecewise or multilayer methods (Tolstoy, 1973; Gossard and Hooke, 1975; Baines, 1998; Smith et al., 2002; Teixeira, 2014; Pütz et al., 2019). These techniques are based on modeling the continuously varying atmosphere as a discrete set of layers, in which the flow's properties

allow for easy computation of the internal wave field. To avoid confusion with the lower and upper layers of the TLM, the layers used by piecewise methods will be called *sublayers* from now on.

### 3.2.1 General principle

The basic principle of piecewise methods is to represent the atmosphere as a discrete number of sublayers (Tolstoy, 1973; Gossard and Hooke, 1975; Baines, 1998). In these sublayers, the atmosphere's properties should vary in a such way that

solutions to the Helmholtz equation are easily found, while still approximating the actual profiles. By increasing the number of sublayers, most aspects of the real flow's behavior can be accurately simulated. A suitable approach is to use sublayers in which $m^2$ has simple profiles, for which analytic solutions can be found. In this work, the atmosphere will be approximated in a piecewise-uniform fashion, so that $m^2$ is piecewise constant, and given in each sublayer by equation 5. The internal wave field in each sublayer is a superposition of exponential functions, corresponding to upwards and downwards traveling waves.

The main advantage of this method is that realistic wavepatterns can be obtained with a relatively small number of sublayers. Within a sublayer, only two degrees of freedom have to be determined. Another advantage compared to other methods such as WKB theory is that piecewise methods can account for wave reflection, although they can not incorporate weakly non-linear effects (Gill, 1982).

As the number of sublayers has to be limited for computational reasons, not all of the atmosphere can be approximated.

An appropriate height $H_n$ has to be chosen, above which the atmosphere is considered uniform, so that $W(z > H_n)$ is an exponential function. In general, the piecewise method can be summarized by the following equations (Tolstoy, 1973; Pütz et al., 2019):

$$m^2(z) \approx \widetilde{m}^2(z) \triangleq \begin{cases} m_0^2, & H < z < H_1 \\ \vdots & \vdots \\ m_j^2, & H_j < z < H_{j+1} \\ \vdots & \vdots \\ m_n^2, & H_n < z \end{cases} \tag{21}$$





$$W(z) \approx \widetilde{W}(z) \triangleq \begin{cases} W_0(z) = W_{0+} \exp\left(\imath m_0 z\right) + W_{0-} \exp\left(-\imath m_0 z\right), & H < z < H_1 \\ \vdots & \vdots \\ W_j(z) = W_{j+} \exp\left(\imath m_j z\right) + W_{j-} \exp\left(-\imath m_j z\right), & H_j < z < H_{j+1} \\ \vdots & \vdots \\ W_n(z) = W_{n+} \exp\left(\imath m_n z\right) + W_{n-} \exp\left(-\imath m_n z\right), & H_n < z \end{cases} \tag{22}$$

where $H_j$ denote the altitudes of the interfaces between sublayers, and $n$ is the number of interfaces between the sublayers. The values for the $2(n+1)$ coefficients $W_{j\pm}$, which determine the internal gravity wave field, depend on additional conditions at the boundaries and sublayer interfaces, which will be discussed in Sect. 3.2.2. The atmospheric variables are evaluated in the middle of each sublayer, so that $\widetilde{m}_j^2 = [m(H_j + \Delta H_j/2)]^2$. This results in second-order convergence when approximating
continuously varying atmospheres, as shown in appendix A. Once the coefficients $W_{j\pm}$ have been determined, equation 20 for the stratification coefficients can be evaluated as:

$$\hat{\Phi} = \frac{\Omega_0}{k^2 + l^2} \left( \Omega_0 \imath m_0 \left(W_{0+} - W_{0-}\right) - \frac{d\Omega_0}{dz} \left(W_{0+} + W_{0-}\right) \right) \tag{23}$$

### 3.2.2 Boundary and sublayer interface conditions

The values of the coefficients $W_{j\pm}$ are determined by the displacement of the inversion layer, the boundary conditions imposed
at the interfaces between the sublayers, and the radiation condition. The stratification coefficients are calculated in Fourier components by evaluating the response to a change in ABL height $\eta_t$. For each wavenumber $k,l$, the relation between $\eta(z) = \hat{\eta}(z) \exp \imath(kx + ly)$ and $W(z)$ is given by (Nappo, 2012):

$$\frac{D}{Dt}\eta(z) = -\imath\Omega\hat{\eta}(z)\exp\imath(kx+ly)$$
$$= W(z)\exp\imath(kx+ly) \tag{24}$$

This leads to a boundary condition for $W_0(H)$:

$$-\imath\Omega_0\hat{\eta}_t = W_{0+}\exp\left(\imath m_0 z\right) + W_{0-}\exp\left(-\imath m_0 z\right) \tag{25}$$

This relation is already incorporated in equation 20. The displacement $\eta$ and the pressure perturbation $p'$ have to be continuous over the interfaces between sublayers (Gossard and Hooke, 1975; Klemp and Lilly, 1975; Gill, 1982; Baines, 1998; Smith et al., 2002; Vosper, 2004). For each interface at $z = H_j$ between the sublayers $j-1$ and $j$, this leads to two conditions that Gossard and Hooke (1975) call the *kinematic* and *dynamic* conditions, respectively:

$$\eta_{j-1}\left(H_j\right) = \eta_j\left(H_j\right) \tag{26}$$

$$p'_{j-1}\left(H_j\right) = p'_j\left(H_j\right) \tag{27}$$





Combining equations 24 and 19 allows the kinematic and dynamic conditions to be written in terms of $W(z)$:

$$\left.\frac{W_{j-1}}{\Omega_{j-1}}\right|_{z=H_j} = \left.\frac{W_j}{\Omega_j}\right|_{z=H_j} \tag{28}$$


$$\left.\Omega_{j-1}\left(\Omega_{j-1}\frac{dW_{j-1}}{dz} - \frac{d\Omega_{j-1}}{dz}W_{j-1}\right)\right|_{z=H_j} = \left.\Omega_j\left(\Omega_j\frac{dW_j}{dz} - \frac{d\Omega_j}{dz}W_j\right)\right|_{z=H_j} \tag{29}$$

Pütz et al. (2019) argue that if the background velocities change continuously with altitude, the kinematic and dynamic interface conditions should be replaced by conditions ensuring that the vertical velocity is continuous:

$$\left.W_{j-1}\right|_{z=H_j} = \left.W_j\right|_{z=H_j} \tag{30}$$


$$\left.\frac{dW_{j-1}}{dz}\right|_{z=H_j} = \left.\frac{dW_j}{dz}\right|_{z=H_j} \tag{31}$$

since otherwise $W$ would be discontinuous if $\Omega$ and its derivative were evaluated at the center of each sublayer, as the use of a constant wavenumber would imply. Indeed, using equations 28 and 29 does not result in convergence to the same solution as solving equation 16 with a simple finite difference solver. However, the physical reasoning behind the kinematic and dynamic

boundary conditions is generally valid, indicating that they should always be used. The difference between the setup of Pütz et al. (2019) and the older literature is solved if in equations 28 and 29 not only $W_{j-1}$ and $W_j$ are evaluated at $z = H_j$, but $\Omega_{j-1}$ and $\Omega_j$ as well. While in the piecewise constant method $m_j^2$ should be taken at the center of each layer and held constant throughout, the background velocities and their derivatives should be evaluated at the interfaces when setting up the interface conditions. If changes in background velocity within layers is taken into account in this way, the kinematic and dynamic

matching conditions will automatically simplify to those proposed by Pütz et al. (2019) when appropriate.

Above the highest sublayer, the atmosphere is assumed to be uniformly stratified. This results in the same situation as discussed in Sect. 2.1, with the upper boundary condition determining the sign of $m$ in the propagating regime. For the propagating and evanescent regimes respectively, the root is chosen so that the radiation condition is satisfied, or that the perturbation dies out with altitude.

Combined with the radiation condition applied at height $H_n$, equations 25, 28, and 29 provide $2(n+1)$ relations, which is enough to solve for $W_{j\pm}$. As the equations are linear, and the kinematic and dynamic equations at each interface only involve the wave fields in the adjacent sublayers, the system of equations for each wavenumber can be solved as a banded matrix of size $2(n+1)$ with a bandwidth of five. Solving the system has a computational complexity of $\mathcal{O}(n)$. The total computation for a single profile of $m^2(z)$ with $n = 2048$, including the building of the matrix, takes roughly 0.65s on a personal laptop with

16GB of RAM and an Intel core i7 2.60GHz. To determine the pressure boundary condition in the TLM, the stratification coefficients for all the different wavenumbers have to be calculated, leading to a total complexity of $\mathcal{O}\left(N_x N_y n\right)$. Finally, we emphasize that all stratification coefficients can be precomputed, as their values do not depend on the solution of the TLM. Once identified, the values of $\hat{\Phi}$ can be used in equation 10 to set the pressure boundary condition in the TLM.





### 3.2.3 Inversion and critical layers

If there are inversion layers in the free atmosphere, these can be modeled by discontinuities in $\bar{\theta}$ corresponding to the inversion strengths $\Delta\bar{\theta}$. This can be incorporated in piecewise methods by adding $g' = \frac{g\Delta\bar{\theta}}{\bar{\theta}}$ to the right-hand side of equation 27 (Baines, 1998).

As the wind can vary, the situation can arise where $\bar{u}_g = -\bar{v}_g k/l$, so that $\Omega = 0$. In a reference frame moving with the wind speed, the frequency of the waves is then zero, and the mean flow is no longer perturbed. It is clear from equation 17 that

this corresponds to a singularity. The height at which this occurs is called a *critical level*, and exactly what happens is hard to predict with linear theory. In general, the wave disappears, and the energy it carried is absorbed by the mean flow (Gossard and Hooke, 1975; Gill, 1982; Baines, 1998). Therefore, critical levels are modelled as fully absorbing sublayers, as is commonly done in literature (Smith et al., 2002; Wells and Vosper, 2010). This is implemented by applying the radiation condition at interface $j$ when $\Omega$ changes sign across interface $j+1$, or becomes zero in sublayer $j+1$. Since there is no difference for the

flow below the critical level between wave energy being absorbed aloft or just radiating outward indefinitely, this is equivalent for our purposes.

### 3.3 Verification

To verify the implementation of the piecewise constant method, it was compared to a second-order finite difference (FD) code with a central difference scheme on various continuously varying background velocities and buoyancy frequencies. The

piecewise method consistently outperformed the FD code, achieving second order convergence as expected through the proof in appendix A, and having small errors even at coarse grids.

We also reproduced results from Wells and Vosper (2010), who calculated the linear gravity response for an idealized atmosphere to a small 2D ridge, described by:

$$h = \frac{h_r}{(x/L_r)^2 + 1} \tag{32}$$

where $h_r = 10$m and $L_r = 10$km. These values were chosen so that the effects of non-linearity would remain small (Wells and Vosper, 2010). The vertical velocity perturbation was computed on a 1000km long domain with 2048 gridpoints, a 1D version of the grid that will be used in Sect. 4, and up to a height of 20km with a grid spacing of 125m, which is comparable to what will be used in Sect. 4.3. The idealized background velocity and Brunt-Väisäla frequency profiles, as well as the gravity wave response, are shown in Figure 1. The contour plots agree fairly well with those obtained by Wells and Vosper (2010), indicating

that the piecewise method is correctly implemented, and suited to this application. This test case will be further discussed in Sect. 4.

Wells and Vosper (2010) also analysed the hydrostatic drag response to the same ridge for atmospheres with a two-layer buoyancy-frequency structure. This is a classic test case, and similar set-ups have been discussed by Gill (1982) and Teixeira and Argain (2020), among others. Wells and Vosper (2010) considered a case where the background wind $\bar{u}(z)$ is constant, and



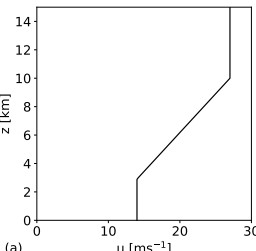 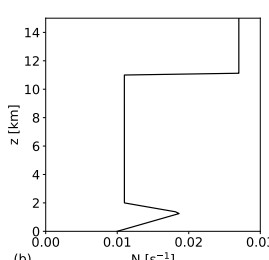 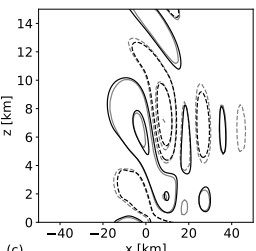

**Figure 1.** Idealized upper atmosphere velocity (a) and Brunt-Väisäla frequency (b) used by Wells and Vosper (2010), and the vertical velocity field as calculated with the piecewise constant method (black lines) and as found by Wells and Vosper (2010) (grey lines) (c). The contour interval is 0.01 m/s, and the solid and dashed lines denote positive and negative values respectively.

one where it is given by:

$$\bar{u}(z) = u_0 + u_1 \sin\left(\frac{\pi z}{2 z_i}\right) \tag{33}$$

where $z_i = 10$km. In the constant wind case, the drag is computed for a range of $u_0$. In the case with vertical wind shear, it is computed for a range of $u_1$, with $u_0$ kept at $5$m/s. In reproducing their results, we used a step profile for the Brunt-Väisäla frequency:

$$N_g(z) = \begin{cases} N_1, & z < z_i \\ N_2, & z > z_i \end{cases} \tag{34}$$

where $N_1 = 0.01\text{s}^{-1}$ and $N_2 = 0.02\text{s}^{-1}$. Our results, obtained on the same grid as above with our method adapted to the hydrostatic regime, and those obtained by Wells and Vosper (2010) are shown in figure 2. Again, the good agreement indicates that the multilayer method performs well.

## 4 Effects of vertically varying wind and stability

By using the piecewise-constant method developed in Sect. 3.2 to evaluate the stratification coefficients, the TLM can now take the variation of the stratification strength and wind speed with altitude into account. The impact of this non-uniformity on the interaction between gravity waves and wind farms is now analysed in three ways. In Sect. 4.1, the impact of atmospheric profile variations on the stratification coefficients is analyzed by comparing $\hat{\Phi}$ for the idealized atmosphere used in Sect. 3.3 to the $\hat{\Phi}$ for a uniform upper atmosphere. The physical phenomenon causing the differences is identified as internal gravity wave
interference.

We further investigate how this influences the interaction between wind farms and gravity waves. To this end, Sect. 4.2 discusses two example cases of ABL flow with uniform upper atmospheres from Allaerts and Meyers (2019). By combining these cases with the atmospheric profiles analyzed in previous sections, we investigate the effect of vertical variations in the




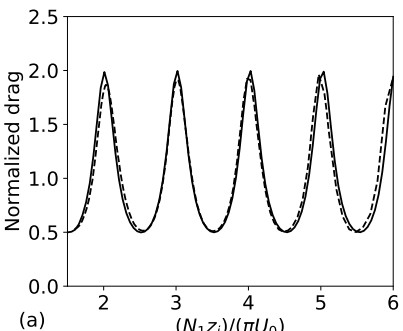
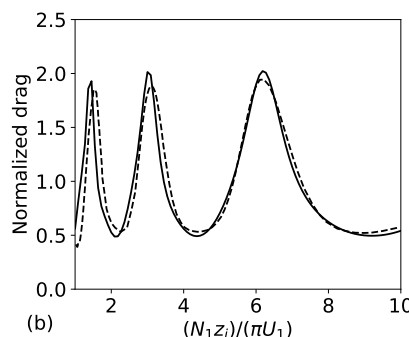

**Figure 2.** Mountain wave drag on a small ridge with a two-layer Brunt-Väisälä frequency profile and constant background wind (a) and vertical wind shear (b), normalized by the drag for constant background profiles. The full lines show our results, in which the wave drag is normalized with the drag for constant background wind $u_0$ and stratification strength $N_1$. The dashed lines show the results of Wells and Vosper (2010).

atmospheric conditions on wind-farm gravity-wave interaction. Finally, Sect. 4.3 presents a case study that estimates the overall
impact of such variations by simulating one year of operation of the Belgian–Dutch offshore wind-farm cluster.

## 4.1 Stratification coefficients

To determine the effects of vertical non-uniformity on the pressure feedback of internal gravity waves, the stratification coefficients are calculated for the upper atmosphere used by Wells and Vosper (2010), shown in figure 1. The same grid as in Sect. 3.3 is used. The coefficients $\hat{\Phi}$ are calculated for both the original non-uniform as well as for a uniform atmosphere. The latter
was obtained by height-averaging the profiles of the velocity and the Brunt-Väisälä frequency, resulting in $U = 20.1\mathrm{m/s}$ and $N_g = 0.013\mathrm{s}^{-1}$. Figure 3 shows the results. Only $k < 1\mathrm{km}^{-1}$ is shown, so that the important details are clearly visible. Since this is a 2D case, the transversal wavenumber $l$ is zero. Comparing the two profiles for $||\hat{\Phi}||$, it is clear that the vertical non-uniformity has a large impact on the stratification coefficients, with two peaks appearing at $k \approx 0.36\mathrm{km}^{-1}$ and $k \approx 0.72\mathrm{km}^{-1}$. We will now show that these changes in the profile of $||\hat{\Phi}||$ are caused by resonance in the free atmosphere.

### 310 4.1.1 Internal wave resonance

While in a uniform atmosphere the wind farm can only trigger waves with an upwards group velocity, changes in the stratification and wind speed can cause waves to reflect. This allows both up- and downgoing waves to propagate throughout the atmosphere. As up- and downgoing internal gravity waves pass through each other, they interfere, potentially causing resonance. The resulting large wave amplitudes can drastically affect the pressure feedback of the waves (Teixeira, 2014). The
ratio $A$ between the mean wave energy at a given wavenumber for atmospheric profile with and without vertical variations is an effective measure for gravity wave resonance, as long as the waves are in the propagating regime (Gill, 1982). It is easily



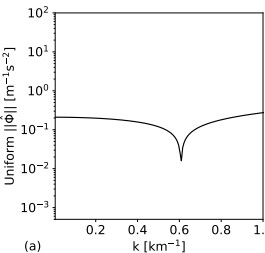 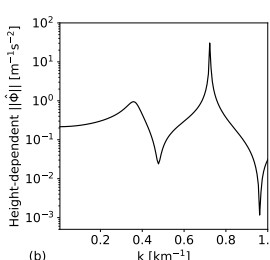 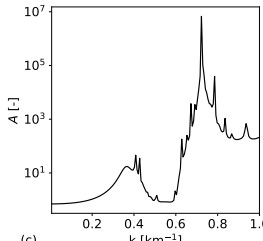

**Figure 3.** The stratification coefficients for a uniform atmosphere with $U = 20.1m/s$ and $N_g = 0.0113s^{-1}$ (a), the stratification coefficients (b) and the gravity wave resonance parameter (see equation 35) (c) for the vertical non-uniform atmosphere of Wells and Vosper (2010) used in Sect. 3.3. Only $k < 1\text{km}^{-1}$ is shown, so that the important details are clearly visible. Since this is a 2D case, the transversal wavenumber $l$ is zero.

evaluated numerically using the following expression:

$$A = \int\limits_{H}^{H_n} e_{HD}(z)dz \cdot \left[\int\limits_{H}^{H_n} e_U(z)dz\right]^{-1}, \qquad (35)$$

where $e$ is the sum of the kinetic and potential energy densities (Gill, 1982):

$$e = \frac{1}{2}\bar{\rho}\left(\overline{u'^2 + v'^2 + w'^2}\right) + \frac{1}{2}g^2\overline{\rho'^2}/\bar{\rho}N_g^2. \qquad (36)$$

For a sublayer in a piecewise-constant method, the integrals are straightforward to determine analytically, allowing $A$ to be evaluated.

Figure 3 shows $A$ for the atmosphere of Wells and Vosper (2010). It is clear that $||\hat{\Phi}||$ and $A$ have a similar profile, with the same peaks occurring at $k \approx 0.36\text{km}^{-1}$ and $k \approx 0.72\text{km}^{-1}$. Above $k = 0.408\text{km}^{-1}$, the gravity waves become evanescent
within some sublayers, leading to oscillations in $A$ that do not correspond to resonant behaviour. Despite this, it's clear from the figure that the profiles of $||\hat{\Phi}||$ and $A$ follow the same pattern, showing that the pressure feedback is largely determined by constructive and destructive interference of the internal gravity waves.

### 4.2 Gravity-wave ABL interaction

We investigate how variations in the wind and stability changes the effects wind-farm operation has on the ABL flow by revising
two example cases used by Allaerts and Meyers (2019). By combining these cases with the non-uniform upper atmosphere used in the previous sections, we identify the mesoscale flow perturbations triggered by the wind farm. One of these changes caused by the variations in the atmospheric conditions — the increase in the appearance and strength of resonant lee waves — is further analyzed.



### 4.2.1 Example cases

To analyze how the changes in the stratification coefficients impacts the flows around wind farms, two flow cases discussed
earlier by Allaerts and Meyers (2019) are analyzed. These example case is simulated once with uniform upper atmospheres,
and once with the upper atmosphere discussed in Sect. 3.3 and 4.1.

The two example cases are set up to have *Froude numbers* of $Fr = 0.9$ and $Fr = 1.1$, with the Froude number given by
(Allaerts and Meyers, 2019):

$$Fr = \frac{\bar{u}_B}{\sqrt{g'H}}, \tag{37}$$

where $\bar{u}_B$ is a velocity scale for the ABL (Allaerts and Meyers, 2019):

$$\bar{u}_B = \left( \frac{H_1}{H} \frac{1}{\bar{u}_1^2} + \frac{H_2}{H} \frac{1}{\bar{u}_2^2} \right)^{-1/2}. \tag{38}$$

The Froude number represents the ratio of the advection terms in the momentum balance to the pressure gradient generated
by the inversion waves (Durran, 1990). It is also a measure for the ratio of the wind speed to the velocity of inversion waves,
and therefore determines whether or not inversion waves can travel upstream from a stationary perturbation. This leads to the
classification of flows as being either subcritical $Fr < 1$, when inversion waves can affect the upstream flow, or supercritical
$Fr > 1$, when they can not (Smith, 2010). One of the example cases of Allaerts and Meyers (2019) is subcritical, with $Fr = 0.9$,
while the other is supercritical, with $Fr = 1.1$.

The example cases are set up as follows. The mean flow in the two layers of the ABL is based on the analytical boundary
layer model of Nieuwstadt (1983), with a cubic eddy viscosity profile $\nu_\tau = \kappa u_* z (1 - z/H)^2$ and the Von Kármán constant
$\kappa = 0.41$ (Allaerts and Meyers, 2019). This model finds a velocity profile given a non-dimensional surface roughness length
$\bar{z}_0 = z_0/H$ and non-dimensional boundary layer height $h_* = H f_c/u_*$. Combined with the stratification parameters $g'$ and
$N_g$, this gives all the inputs required for the TLM. Allaerts and Meyers (2019) presume a conventionally neutral boundary
layer, determining these stratification parameters with the non-dimensional groups $N_g/f_c$ and $g'H/Au_*^2$, where $A = 500$ is an
empirical constant (Csanady, 1974; Tjernström and Smedman, 1993). Since we want to analyze the impact vertical variations
in the upper atmospheric profile have on the flow, the original inputs used by Allaerts and Meyers (2019) are altered so that the
upper atmosphere corresponds to the one analyzed in Sect. 4.1. The correct geostrophic wind and Brunt-Väisäla frequency are
obtained by modifying $u_*$ and $N_g/f_c$ respectively, and $g'H/Au_*^2$ is changed so that the final results still have Froude numbers
of 0.9 and 1.1. $H$ is increased so that $h_*$ and $\bar{z}_0$ remain the same. The inputs are summarized in table 1. Finally, the flow in the
lower layer is aligned with the $x$-direction.

Table 2 gives an overview of the wind-farm configuration used by Allaerts and Meyers (2019), which is chosen to be
comparable to the Belgian–Dutch wind-farm cluster in area and installed capacity. The turbines are placed in a staggered
pattern with respect to the $x$-direction, and the relative spacing is equal in the $x$- and $y$-directions. The simulations were
performed on a 1000 km by 400 km grid, with 2000 by 800 gridpoints, which is the same as the grid spacing as was used in
Sect. 3.3. For the vertically non-uniform simulation, the same vertical grid as in Sect. 3.3 and 4.1 is used.





**Table 1.** Flow parameters of the sub- and supercritical flow cases based on those used by Allaerts and Meyers (2019). The atmospheric state corresponds to a $P_N$ of 1.1, and $Fr$ of 0.9 and 1.1.

| Formula | Definition | Value | Value used by Allaerts and Meyers (2019) |
|---------|------------|-------|-------------------------------------------|
| $h_*$ | Non-dimensional boundary layer height | 0.15 | 0.15 |
| $\bar{z}_0$ | Non-dimensional surface roughness length | $10^{-4}$ | $10^{-4}$ |
| $g'H/Au_*^2$ | Inversion parameter (subcritical) | 1.07 | 0.69 |
| | Inversion parameter (supercritical) | 0.71 | 1.04 |
| $N/f_c$ | Brunt-Väisäla frequency to Coriolis parameter | 113 | 58 |

**Table 2.** Wind farm configuration of the reference flow cases, as analyzed by Allaerts and Meyers (2019).

| Configuration | Value |
|---------------|-------|
| Wind-farm length | 20 km |
| Wind-farm width | 30 km |
| # turbine rows | 18 |
| # turbine columns | 27 |
| Rotor diameter | 154 m |
| Thrust coefficient | 0.8 |
| Relative turbine spacing | 7.21 |

The results of the uniform and non-uniform simulations are shown in figures 4 and 5, respectively. When comparing the results, it's clear that the mesoscale disturbances are much larger in the non-uniform cases, with a stronger reduction of the farm inflow velocity. The farm's effects also spread out in the transversal directions, as V-shaped patterns appear over large distances. Finally, the wind farm seems to trigger strong lee waves in its wake. These waves also appear in the subcritical reference case, as analyzed by Allaerts and Meyers (2019), although they are weaker there.

#### 4.2.2 Resonant lee waves in the ABL

We further analyse the lee waves that appear in figure 5, and show that they are associated in this case to internal wave reflection. We follow the ideas developed by Allaerts and Meyers (2019) by analyzing the equation for the total displacement $\eta_t$. When the wind in the ABL is aligned with the $x$-axis ($\bar{v}_1 = \bar{v}_2 = 0$), that equation corresponds to (Allaerts and Meyers, 2019):

$$\left(-1 + Fr^{-2} + P_N^{-1}\frac{\Phi}{GN_g}\right) * \frac{\partial^2 \eta_t}{\partial x^2} + \left(Fr^{-2} + P_N^{-1}\frac{\Phi}{GN_g}\right) * \frac{\partial^2 \eta_t}{\partial y^2} = \frac{H_1}{\bar{u}_1^2}\nabla \cdot RHS_1 + \frac{H_2}{\bar{u}_2^2}\nabla \cdot RHS_2, \tag{39}$$



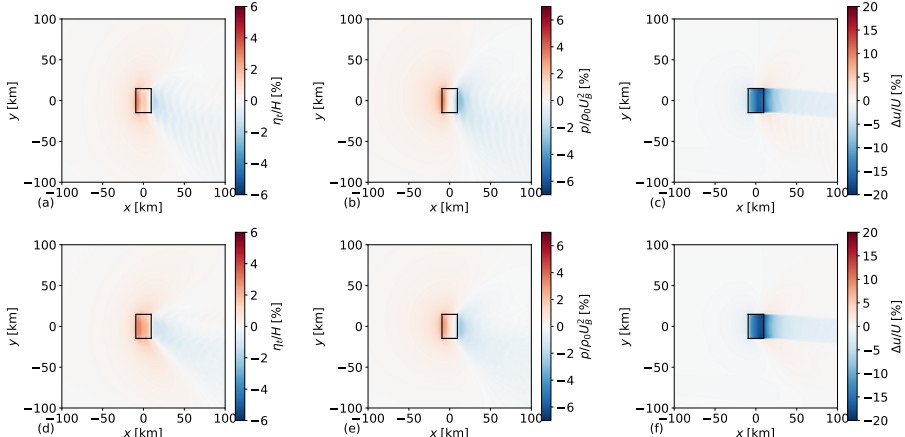

**Figure 4.** Planform view of the inversion layer displacement $\eta_t$ (left), pressure perturbation $p'$ (middle), and velocity reduction in the lower layer $\boldsymbol{u}'$ (right) in the reference uniform sub- (top) and supercritical (bottom) flow cases. The wind-farm region is indicated by the black rectangles.

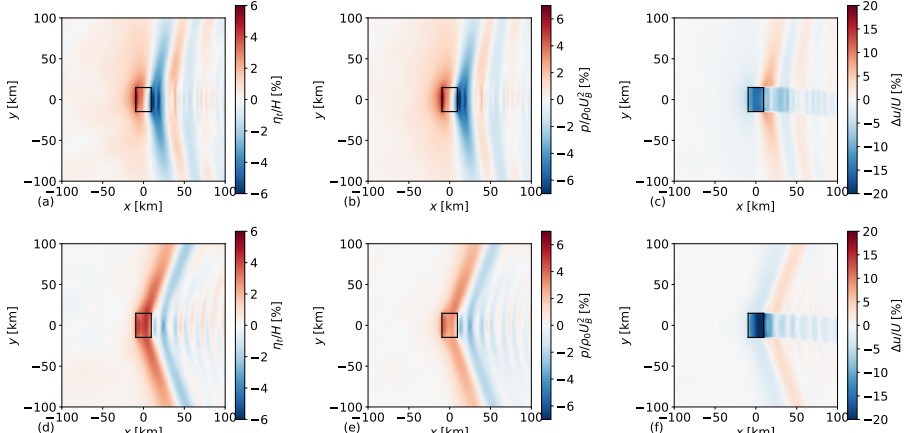

**Figure 5.** Planform view of the inversion layer displacement $\eta_t$ (left), pressure perturbation $p'$ (middle), and velocity reduction in the lower layer $\boldsymbol{u}'$ (right) in the reference vertically non-uniform sub- (top) and supercritical (bottom) flow cases. The wind-farm region is indicated by the black rectangles.

where $G$ is the geostrophic wind velocity. Furthermore, $RHS_1$ and $RHS_2$ are the right-hand sides of the equations 6 and 7, respectively, and only depend on $\eta_t$ through the turbulent viscosity, which is a relatively weak effect (Allaerts and Meyers, 2019). As discussed before, the Froude number $Fr$ indicates the strength of the inversion waves. The second non-dimensional group $P_N$ governs the pressure induced by internal gravity waves, and is given by:

$$P_N = \frac{\bar{u}_B^2}{G N_g H}.$$
(40)





It reflects the fact that the pressure induced by a vertical displacement scales linearly with $GN_g$ in the hydrostatic regime. It is then clear that the left-hand side of equation 39 represents the forcing induced by flow advection $\left(-\frac{\partial^2 \eta_t}{\partial x^2}\right)$ and the corresponding gravity waves $\left(\left(Fr^{-2} + P_N^{-1}\Phi/GN_g*\right)\nabla_H^2\eta_t\right)$. These are balanced by the right-hand side of the equation, which represents the other terms in the momentum equations. If the advection terms and the pressure contributions from the

gravity waves balance each other, the left hand side of equation 39 becomes zero. This corresponds to a resonant state, as $\eta_t$ can be non-zero without external forcing, which explains the lee waves that appear in figure 5.

The left-hand side of equation 39 is easily expressed in Fourier components, leading to the definition of the two-dimensional lee-wave resonance parameter $R$:

$$R = (\cos\lambda)^2 - Fr^{-2} - \frac{H}{\bar{u}_B^2}\hat{\Phi}, \tag{41}$$

where $\lambda$ is the angle the horizontal wavevector makes with the $k$-axis, so that $(\cos\lambda)^2 = k^2/(k^2 + l^2)$. The parameter $R$ is a non-dimensional parameter indicating the flow's resistance to the occurence of two-dimensional lee-wave resonance. If $R = 0$, the pressure perturbations induced by $\eta_t \neq 0$ and the accompanying gravity waves balance the advection terms, and resonant lee waves appear. Equation 41, in combination with equation 4, shows that for uniform upper atmospheres, this type of resonance can only take place if the internal gravity waves are evanescent, as $\hat{\Phi}$ has to be real for $R$ to be zero. Physically, this corresponds

to propagating waves not being able to trap the perturbation energy below the capping inversion (Vosper, 2004). In contrast, the wave reflection in vertically non-uniform atmospheres can cause part of the wave energy to be trapped by being reflected back. This corresponds to the stratification coefficients as given by equation 23 potentially having a real component, even in the propagating regime. The additional constraint found by Allaerts and Meyers (2019), that the flow has to be subcritical, is also not necessary when the waves can be reflected.

In order to apply this theory to the cases from Sect. 4.2.1, figure 6 shows the wavenumber spectra for $\eta_t$ and $R$ for the supercritical case. Figure 6 shows that low values of $||R||$ correspond to high values of $||\hat{\eta}_t||$, indicating resonant lee waves in the ABL. The variations in wind and stability in the upper atmosphere can cause $||R||$ to decrease several orders of magnitude in the propagating wave regime. This happens when the internal gravity wave interference is destructive, which lowers the vertical energy flux of the waves considerably. This keeps the perturbation energy contained in the lower atmosphere, leading to low

$||R||$, which for uniform atmospheres only happes in the evanescent wave regime (Vosper, 2004; Allaerts and Meyers, 2019). In contrast, constructive interference can lead to very large values for $||\hat{\Phi}||$, and thus for $||R||$. Furthermore, when $||R|| \gg 0$, a large imbalance exists between the advection and pressure forces in the ABL. Changes in ABL height of the wavelengths at which this occurs cannot exist in an equilibrium system without external forcing. This is clearly visible in figure 6, as high values of $||R||$ correspond to low values of $||\hat{\eta}_t||$. Internal gravity wave resonance thus prohibits large displacements of the

inversion layer at the wavelengths at which it occurs, by creating large pressure gradients that can not be counteracted by the flow acceleration.

While the above analysis explains how vertical variations in the atmospheric profile change the interaction between internal gravity waves and the ABL flow, it does not offer insight on how to predict the resulting impact on wind farm performance. It is not clear what parameters could describe this. Extensive research has been done on internal gravity wave resonance, with most




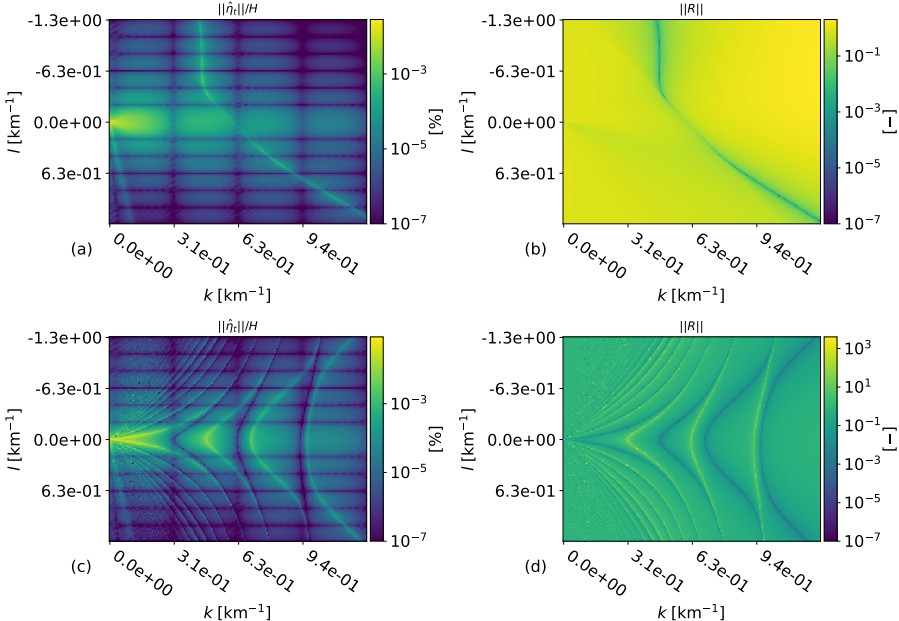

**Figure 6.** $||\hat{\eta}_t||/H$ (left) and $||R||$ (right) for the reference supercritical flow case with a uniform (top) and a non-uniform free atmosphere (bottom). Only the wavenumbers $||k|| \leq 1.26\,\mathrm{km}^{-1}$ and $||l|| \leq 1.26\,\mathrm{km}^{-1}$ are shown, so that the important details are clearly visible. The left column shows parts of the Fourier transforms of figures 4 and 5.

of it focusing on flow around topographies. However, this has to be used with caution, as the overall flow is then analyzed in the context of mountain wave drag, where the height displacement is given by the shape of the terrain (Teixeira, 2014). In contrast, the wind-farm forcing leads to this displacement through its interaction with the ABL. Additionally, it is itself influenced by the mesoscale effects it triggers, leading to an additional feedback loop. Parameters successfully describing internal wave resonance may therefore not be able to predict how vertical non-uniformity will impact the interaction with wind-farms.

## 4.3 Overall impact

To determine the impact of varying wind speeds and stability on wind-farm energy production, we follow the approach of Allaerts et al. (2018) by simulating one year of wind-farm operation of the Belgian-Dutch wind-farm cluster. This analysis is performed with both uniform and non-uniform upper atmospheres, and the two are compared. The TLM input is based on ERA5 reanalysis data of the year 2016, which is available at hourly frequency, resulting 8784 flow cases. Like Allaerts et al. (2018),

we use data from the grid point nearest to the wind-farm cluster, at 51.6N 3.0E, and the same approach in determining the TLM input from the atmospheric data. We further assume that all turbines are DTU 10 MW reference turbines, a commonly analyzed model for which the power curve is readily available (Bortolotti et al., 2019), and place them within the same trapezoidal area as Allaerts et al. (2018). To ensure that we obtain the same total installed capacity of 3.8GW, we scale the number of turbines



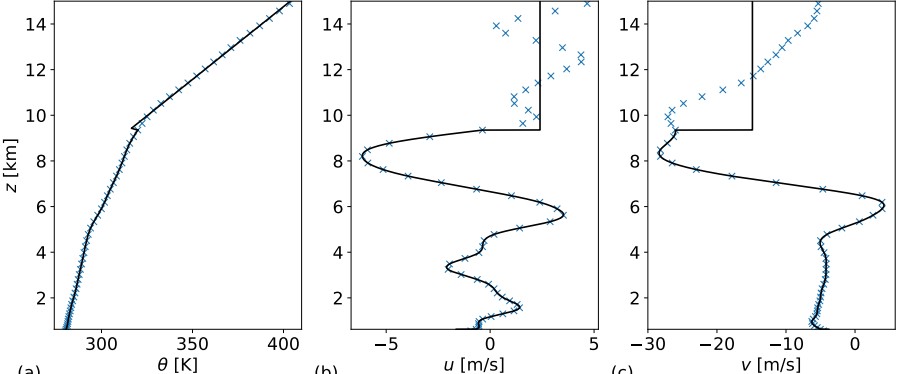

**Figure 7.** The atmospheric profiles of potential temperature (a), velocity in the $x$- (b) and $y$-direction (c). The blue crosses are the ERA5 data, and the black lines are the inputs to the TLM, showing the piecewise-constant approximation made in the discretization.

and the turbine spacing from 475 and 7.36 to 380 and 8.23 respectively. The thrust coefficient of the turbines is set based on
the height-averaged background velocity in the wind-farm layer of the TLM.

The simulations were performed on a 2000 by 2000 grid, on a 1000km by 1000km domain, the same grid density as in the previous sections. Variations in the atmospheric profiles were taken into account up to the tropopause. Within the troposphere, the potential temperature and velocity profiles were modelled as first and third order splines, respectively, through the ERA5 data points, around which the sublayers were spaced as well, resulting in 30 to 50 sublayers for each case. The
tropopause altitude and the stratification in the stratosphere were determined with a 2-line, piecewise linear regression fit on the temperature profiles between the ABL height and 15km. The velocity in the stratosphere was then determined by height-averaging the profile from the tropopause up to 15km. As an example, figure 7 shows the profiles of $\theta$, $u$, and $v$ for the upper atmosphere of 12am May 1st, 2016.

From the 8746 cases, we only use those where the atmosphere is statically stable at every altitude in the free atmosphere.
Additionally, the cases without capping inversion, (cf. earlier discussion following equation 10 in Sect. 2.2), or with a capping inversion situated lower than twice the turbine hub height were left out, leaving 3890 cases. This filtering was necessary, as for the removed cases the assumptions made in the derivation of the model are not valid, as discussed in Sect. 2.2. The results of the simulations are shown in figure 8, and summarized in table 3. On average, the difference between the results with non-uniform and uniform upper atmospheres is small. Despite this however, the impact on individual cases is often significant. In 16.5%
of the analyzed cases, the difference between the uniform and the non-uniform inflow perturbation was more than 30% of the uniform perturbation case. We therefore conclude that vertical variations of the upper atmospheric profiles are important to take into account when analyzing individual flow cases.

Figure 9 shows that the differences between the simulations seem to be independent from parameters that were good predictors of the TLM's behaviour in previous studies, such as $Fr$ and $P_N$. As mentioned in Sect. 4.2.2, known parameters from
internal gravity wave theory may not be suited to predict the impact on wind farm operation.



**Table 3.** Average perturbations over all the analyzed flow cases for both uniform and non-uniform upper atmospheres.

| Average over analyzed flow cases | Uniform | Non-uniform |
|---|---|---|
| Maximum $\eta_t/H$ | 10.59% | 10.61% |
| Inflow $\Delta u_1/\bar{u}_1$ | 4.05% | 4.08% |

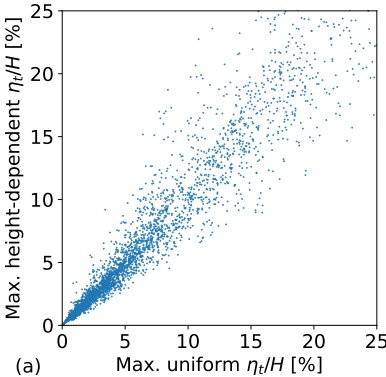
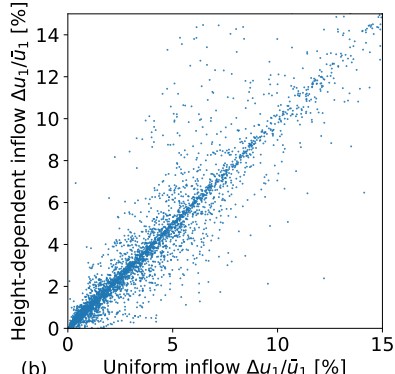

(a)  Max. uniform $\eta_t/H$ [%]

(b)  Uniform inflow $\Delta u_1/\bar{u}_1$ [%]

**Figure 8.** The maximum capping inversion displacement (a) and the inflow velocity perturbation (b) for all analyzed cases with both uniform ($x$-axis) and non-uniform ($y$-axis) upper atmospheres.

## 5 Conclusions

The goal of this study was to extend the applicability of a wind-farm gravity-wave model to vertically non-uniform free atmospheres. This was done by changing the expressions for the stratification coefficients $\hat{\Phi}$ to results derived from the internal wave equation for general stratified flows. By applying the well-known piecewise method with large numbers of sublayers,
general stratification and velocity profiles can be incorporated into the model.

The effects of the variations in background wind and stability were studied by analyzing how free atmospheric wave reflection influences the wave pressure feedback, the ABL flow, and overall wind farm performance. Firstly, the stratification coefficients for the idealized atmosphere used by Wells and Vosper (2010) were compared to those for a uniform atmosphere based on it. The differences were found to be caused by constructive and destructive interference of the internal gravity waves in the free atmosphere. In a second step, this vertically non-uniform atmosphere was combined with the flow cases used by Al-
laerts and Meyers (2019). An analysis of the appearance of resonant lee waves led to a qualitative understanding of the vertical non-uniformity's effects on the interaction between gravity waves and the ABL flow. Due to destructive internal wave interference, resonant lee waves can appear. On the other hand, internal gravity wave resonance dampens inversion layer displacement for the wavelengths at which it occurs.



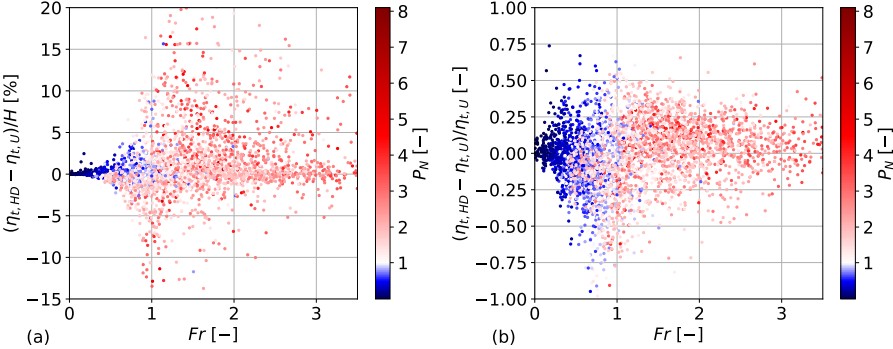

**Figure 9.** The difference between the maximum capping inversion displacements in the non-uniform and uniform simulations as normalized by the ABL height (a) and the displacement in the uniform simulation (b), plotted against the parameters $Fr$ and $P_N$. The lack of a clear trend in both figures indicates that $Fr$ and $P_N$ are not related to the impact of the vertical non-uniformity.

Finally, the extended TLM was used to simulate one year of operation of the Belgian-Dutch wind-farm cluster, repeating a similar analysis by Allaerts et al. (2018), in order to determine overall impact of variations in the atmospheric profiles on the interaction between the ABL flow and the gravity waves to be determined. While this impact was found to be small when averaged out over all flow cases, on individual flow cases it is often of the same order of magnitude as the total flow perturbation. In 16.5% of the analyzed flows, the variations in the atmospheric profiles caused a relative difference in the upstream velocity

reduction of more than 30%. It is unclear how the effect of this non-uniformity for individual flow cases can be predicted, since it does not simply scale with $Fr$ or $P_N$.

The results of this study show that vertical atmospheric non-uniformity could play a major role in the interaction between wind farms and gravity waves. This suggests that variations with altitude of the free atmosphere's wind and stability should be taken into account when simulating wind-farm operation in specific atmospheric conditions, and may be important for

the optimization of turbine control in the future (see, eg. Lanzilao and Meyers (2021b)). In the future, we foresee further improvements of the TLM, among others, including hydrodynamic effects in the boundary layer, and upgrading the wake model to include the improved wake merging model by Lanzilao and Meyers (2021a). Next to that, we plan further validation against detailed large-eddy simulations (similar to Allaerts and Meyers (2019)), and data from operational wind farms.

*Code and data availability.*  The code used for the simulations and the raw data of the simulation results can be provided by contacting the

corresponding author. The code used for the simulations is written in Python.





## Appendix A: Convergence analysis

Adding more sublayers leads to a better approximation of the actual profiles of atmospheric variables. This would then also result in a better approximation of $W(z)$. To estimate the rate of convergence, $z_*$ is defined as the altitude where $\frac{\partial \widetilde{W}}{\partial z}(z_*) = \frac{\partial W}{\partial z}(z_*)$. If $n$ is sufficiently large, such an altitude exists in each sublayer. The error $E = W - \widetilde{W}$ can then be written as a Taylor expansion around $z_*$:

$$
\begin{aligned}
E(z) &= W(z) - \widetilde{W}(z) \\
&= W(z_*) + \frac{\partial W}{\partial z}(z_*)\Delta z + \frac{1}{2}\frac{d^2 W}{dz^2}(z_*)\Delta z^2 \\
&\quad - \widetilde{W}(z_*) - \frac{d\widetilde{W}}{dz}(z_*)\Delta z - \frac{1}{2}\frac{d^2\widetilde{W}}{dz^2}(z_*)\Delta z^2 + \dots \\
&= W(z_*)\left(1 - \frac{1}{2}m^2(z_*)\Delta z^2\right) - \widetilde{W}(z_*)\left(1 - \frac{1}{2}\widetilde{m}^2(z_*)\Delta z^2\right) + \dots
\end{aligned}
\tag{A1}
$$

Using the above result, the change in error over the $j$th sublayer $\Delta_j E$ can be written as:

$$
\begin{aligned}
\Delta_j E &= E(H_{j+1}) - E(H_j) \\
&= W(z_*)\left(1 - \frac{1}{2}m^2(z_*)\Delta z_{j+1}^2\right) - \widetilde{W}(z_*)\left(1 - \frac{1}{2}\widetilde{m}^2(z_*)\Delta z_{j+1}^2\right) \\
&\quad - W(z_*)\left(1 - \frac{1}{2}m^2(z_*)\Delta z_j^2\right) + \widetilde{W}(z_*)\left(1 - \frac{1}{2}\widetilde{m}^2(z_*)\Delta z_j^2\right) + \dots \\
&= \frac{1}{2}m^2(z_*)\left(\Delta z_j^2 - \Delta z_{j+1}^2\right) - \frac{1}{2}\widetilde{m}^2(z_*)\left(\Delta z_j^2 - \Delta z_{j+1}^2\right) + \dots
\end{aligned}
\tag{A2}
$$

Because $\Delta z_{j+1} + \Delta z_j = \Delta_j H$, with $\Delta_j H$ the thickness of the $j$th sublayer, a maximum value for $|\Delta E_j|$ is given by:

$$
|\Delta_j E| \leq \frac{1}{2}\left|m^2(z_*) - \widetilde{m}^2(z_*)\right|\Delta_j H^2 + \dots
\tag{A3}
$$

If $m^2 = \widetilde{m}^2$ somewhere in the sublayer, substituting a Taylor expansion for $m^2$ then directly leads to:

$$
|\Delta_j E| \leq \frac{1}{2}\left|\frac{dm^2}{dz}(z_*)\right|\Delta_j H^3 + \dots
\tag{A4}
$$

From this analysis, it is clear that the piecewise-constant method has the best result when the distance to $z_*$ is minimized for all $z$. Therefore, when approximating a general continuously varying atmosphere, sublayers should be evenly spaced, and the atmospheric state should be evaluated halfway through each sublayer when calculating $\widetilde{m}_j^2$. In that case, the maximum error can be expected to scale with $n|\Delta_j E|$, and $\Delta_j H \sim n^{-1}$, resulting in a second order rate of convergence, as also found with a different derivation by (Pütz et al., 2019). This is confirmed by comparison with a finite difference solver.

It is notable that the above derivation is not limited to piecewise-constant methods, but is valid for general piecewise methods. Therefore, as long as $\widetilde{W}(z)$ is not set up so that $\widetilde{W}(z_*) = W(z_*)$, the convergence rate will remain second order. For example, we also developed a piecewise-linear methods using Airy functions instead of exponential functions, that did not outperform the piecewise-constant one. This is because when linearly approximating a general function $m^2(z)$, $\widetilde{W}(z_*)$ and $W(z_*)$ will still not coincide. As a result, there is no improvement for general atmospheric profiles.



*Author contributions.*  KD, LL, NVL, and JM jointly derived how to apply the piecewise method to the TLM, and set up the verification and simulation studies. KD performed code implementations and carried out the simulations. SJ provided the optimization tool used to determine the altitude of the tropopause. KD, LL, SJ, NVL, and JM jointly wrote the manuscript.

*Competing interests.*  The authors declare that they have no conflicts of interest.

*Acknowledgements.*  The authors acknowledge support from the project FREEWIND, funded by the Energy Transition Fund of the Belgian Federal Public Service for Economy, SMEs, and Energy (FOD Economie, K.M.O., Middenstand en Energie). The computational resources and services in this work were provided by the VSC (Flemish Supercomputer Center), funded by the Research Foundation Flanders (FWO) and the Flemish Government department EWI.



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
