# Peer review of "Including realistic upper atmospheres in a wind-farm gravity-wave model"

_Wind Energy Science, 2021_

## Author Comment (AC1)

**1 Response to comments of Referee #1**

- The article is very well written and discusses a problem with a well-established methodology by subdiving the perturbation equations into several layers whene the analytical solution is found, reducing the differential problem to an algebraic linear system.
  We would like to thank the referee for the kind words, and for the helpful feedback.

- 1. Section 2.1 and 2.2 do not have any graphical schematic to help the reader to understand the layers subdivision. I think that adding those (at least for one section) will facilitate the understanding
  We have added a figure from Allaerts and Meyers (2019) to clarify the structure of the TLM, and referenced it at line 101:

  "... For this reason, the lower layer is also called the wind-farm layer. *The resulting approximation of the ABL is visualized in figure 1.* The flow in the two layers is governed by ..."

[Figure]

Figure 1: Schematic representation of the three-layer model. Figure from D. Allaerts and J. Meyers, Journal of Fluid Mechanics, 862, 990-1028 (2019).

- 2. Equation 14. The total derivative operator is undefined. The authors are also focusing on stationary waves, right?
  We indeed only analyse steady-state flows. To clarify this, we have added the definition of the total derivative where it is first used, at line 152:

  "where $\frac{D}{Dt}$ is the intertial derivative, which for steady systems simplifies to $\frac{D}{Dt} = \bar{u}_i \frac{\partial}{\partial x_i}$, and where $w'$ is the vertical velocity perturbation of the wave."

- 3. Equation 20. Are the derivatives evaluated at Z=H?
  Since $\Phi$ describes the relation between pressure and displacement at the top of the ABL, they are indeed defined there. To make this clear, we have changed equation 20 to:

$$\hat{\Phi} = \left[ \frac{\Omega}{k^2 + l^2} \left( \Omega \frac{dW}{dz} - \frac{d\Omega}{dz} W \right) \right] \Bigg|_{z=H}$$

- 4. The method described by equation 20 is very common in acoustics (see the book of Salomons, Computational Atmospheric Acoustics about the FFP method)
  We thank the referee for pointing this out to us. We added the following sentence to the section where the piecewise method is introduced, on line 175:

  "*This approach is also commonly used in acoustics, where it is called the FFP method (Salomons, 2001).*"

- 5. around line 245: since the authors give importance to the computational time, it is worth to state what solver was used to solve the banded matrix? Was that the native numpy routines or did they use a home-made algorithm?
  We agree with the referee that this is important to add. We changed line 243 to:

  "*The total computation for a single profile of $m^2(z)$ with $n = 100$, including the building of the matrix, takes roughly $0.4$s on a personal laptop with 16GB of RAM and an Intel core i7 2.60GHz, using the standard numpy solve routine sped-up with the numba package (Lam et al., 2015; Virtanen et al., 2020).*"

  We also changed the number of sublayers used in this example calculation to be more representative of the cases analysed in section 3.3 and 4.

- 6. Line 254. I would replace frequency with wavenumber since frequency is more related to temporal variations, while your method is for stationary waves. This applies to the entire manuscript
  We agree that our phrasing is unclear, but disagree with the referee's proposed change. Since $\Omega$ is the convective derivative operator in Fourier space, it denotes the temporal frequency the flow experiences for a given wavenumber. Therefore, we changed the wording throughout the paper to describe $\Omega$ as the *intrinsic* frequency.

- 7. Line 289. The agreement is qualitatively well but not perfect. How can one improve the agreement? By adding more layers? or there is a limitation in the original data from Wells and Vosper?
  We find that using a vertical spacing of 100m slightly improved results, as the altitude of the change in Brunt-Vaisala frequency then exactly corresponds to a sublayer interface. However, aside from this, using up to ten times finer grids did not change anything. Furthermore, earlier work by Gill (1982) and Leutbecher (2001) for the case of constant wind velocity agrees with our results.

  To address this in the paper, we have updated figure 2 to now include the results of Leutbecher (2001) for the case with uniform velocity (left panel). This is referenced in the text on line 286 as follows:

  "*Our results, obtained on the same grid as above with our method adapted to the hydrostatic regime, and those obtained by Wells and Vosper (2010), and Leutbecher (2001) for the constant wind case, are shown in figure 2.*"

- There are some typos here and there. I have found two at page 5 at rows 2 and 6 where coefficients and inversions should be singular.
  We thank the referee for finding these. The typo *inversions* has been corrected to *inversion*. The stratification coefficients should be plural however, as there are different coefficients for all the wavenumbers.

[Figure]

[Figure]

Figure 2: Mountain wave drag on a small ridge with a two-layer Brunt-Väisäla frequency profile and constant background wind (a) and vertical wind shear (b), normalized by the drag for constant background profiles. The black lines with squares show our results, in which the wave drag is normalized with the drag for constant background wind $u_0$ and stratification strength $N_1$. The blue lines with circles show the results of Wells and Vosper (2010), and the red line with triangles in the left figure shows the results of Leutbecher (2001).

---

## Author Comment (AC2)

**1   Response to comments of Referee #2**

We thank the referee for their useful comments and feedback. We have carefully incorporated them, as the numbered points discuss in more detail below.

1. Because of the complexity of the tropospheric structure, they do not seem to have isolated the causal relationship between wind response and particular profile features. This random relationship is illustrated in Figure 8 and 9. This is a little disappointing.
   We have indeed not found this, and leave determining such a causal relationship to future studies. We go into more detail in the response to point 2.

2. On the above point, it might be worth checking the following idea. The impact of profile details on the lower troposphere is probably caused either by the way the waves are launched ( the low level N and U) or by the way that waves are reflected downwards. If the latter is true, then their idealized abrupt tropopause might be important as it is probably the main reflector (Fig 2?). An early paper by Klemp and Lilly (1975, referenced here), tried to explain severe downslope wind based on a tuning related to tropopause reflection. If this is the case, one can define another "Froude number" using the critical speeds for deep wave resonance of this type. Keep in mind however, that tropopause reflection is probably overdone in this model due the assumed sharp tropopause.
   We would like to thank the referee for their interesting suggestions. We will provide an overview of our attempts to find a predictive parameter for the influence of vertical variations.
   Klemp and Lilly (1975) study an atmospheric model very similar to ours, with a CNBL capped by an inversion layer, and a two-layer buoyancy structure above. In doing so, they find that the velocity response to a hill scales with a factor they call $c_1$ (equation 16). Their model is 1D, so in our analysis we used the total velocity magnitude in their expressions. However, we found no correlation between $\eta_t/H$ and $u'/\bar{u}_1$. For the inflow velocity perturbation, this is illustrated here: Teixeira et al. (2013) also analyzed two-layer atmospheres

[Figure]

[Figure]

Figure 1: The parameter $|c_1|$ and the inflow velocity perturbation $u'/\bar{u}_1$ for the uniform (left) and non-uniform (right) ERA5 cases. There is no clear relation between the datasets.

which could potentially capture the effect of the stratosphere. They looked into the influence of the Scorer parameters in both layers and the altitude of the tropopause through the parameters $l_1 H/\pi$ and $l_1/l_2$, but restricted their analysis to cases where $l_1 < l_2$, so that wave reflection would occur. This restriction leaves around 69% of the analyzed ERA5 cases. However, for these cases, we do not find a relation between the ABL flow perturbations and $l_1 H/\pi$ or $l_1/l_2$.
One limitation of both studies mentioned so far is that they only consider one-directional flow. Teixeira et al. (2008, *Mountain Waves in Two-Layer Sheared Flows: Critical-Level Effects, Wave Reflection, and Drag Enhancement*, Journal of the Atmospheric Sciences) found that directional shear has a large impact on ground-level perturbations through the

presence of critical levels. As we took the full complexity of the ERA5 profiles into account up to the tropopause, such directional shear effects could explain the why the parameter of Klemp and Lilly (1975) don't predict the impact of the vertical variations. Future work could explore this further.

Aside from parameters from literature, we also investigated whether $N_{strat}/N_{trop}$ and $||\vec{u}_{strat}||$ $/||\vec{u}_{trop}||$ correlated with the TLM behavior. However, both of these parameters turned out to be independent from it.

Finally, we tried to see if the pressure feedback for a sine wave with spatial scales corresponding to the wind farm's dimensions could predict the size of the flow perturbations. The magnitude of this pressure is straightforward to compute as the sum of the stratification coefficients corresponding to the wavenumbers $\vec{\kappa} = (\pm 2\pi/L_x, \pm 2\pi/Ly)$. Unfortunately, this also did not correlate to the ABL displacement or velocity perturbations.

Within the paper, we modified the final paragraph of section 4.3 to mention these additional investigations, and to give a clearer suggestion for future work:

"*Figure 9 shows that the differences between the simulations seem to be independent from parameters that were good predictors of the TLM's behaviour in previous studies, such as $Fr$ and $P_N$. We also investigated parameters that were found to correlate well with mountain wave drag in vertically non-uniform atmospheres, such as $c_1$ as found by Klemp and Lilly (1975, eq. 16), but could not identify a meaningful correlation in our case. Directional shear effects, such as those investigated by Teixeira et al. (2008), might explain the discrepancy.*"

3. One problem is that the authors discuss a large number of different model runs but the prrofiles are imprecisely described. I found it difficult to know the properties of each run. The problem begins with the Fig 1 where these plots do not match the equations just below (33 and 34). The problem gets worse from there as the reference to different wind and stability profiles are too casual and imprecise. This problem must be fixed for reader to follow the logic of the paper. Perhaps a table of run characteristics would help.
   We added table 1, and reference it at the end of Sect. 3.3 to clear up any confusion:

Table 1: An overview of the different upper atmospheric flow profiles used for verification. The upper atmosphere set up by Wells and Vosper (2010) is also used in Sect. 4.1 and 4.2.

| Upper atmospheric profiles | $u(z)$ | $N(z)$ |
| --- | --- | --- |
| Wells and Vosper (2010) | Figure 2, left | Figure 2, middle |
| Two-layer Brunt-Väisälä frequency, constant wind | Constant | Eq. 34 |
| Two-layer Brunt-Väisälä frequency, varying wind | Eq. 33 | Eq. 34 |

4. I am not sure I see the point of figures 4 and 5. I think they are trying to show the impact of the Froude number based on the inversion strength (g'). It seems from these plots that that Froude number makes little difference. I kind of expected this. When tropospheric stability (N) is very small the inversion Froude number makes of big difference but for realistic values of N, the effect of the inversion is much less. There results in Fig 4 and 5 just seem to verify this general property. The properties aloft are more important than the inversion (g').
   We agree with this assessment. We therefore leave out the subcritical case from the analysis, and only show and compare the uniform and vertically varying supercritical case. Section 4.2 has been rewritten, but the results of the analysis remain the same.

5. In broad terms I think this paper is valuable and significant as it points out that variable tropospheric wind and stability profiles significantly impact wind farm disturbance patterns.

We would like to thank the referee for the kind words, and for their helpful suggestions.

6. Minor points:

   (a) A little more explanation of Fig. 3 would be helpful. Why so many peaks?
   We added the explanation that was given for this on line 324 within the figure description:
   "*Above $k = 0.408 km^{-1}$, the gravity waves become evanescent within some sublayers, leading to oscillations in $A$ that do not correspond to resonant behaviour.*"

   (b) Line 122 Why does hydrostatic require the inversion?
   The hydrostatic assumption requires that pressure perturbations within the ABL are small compared to pressure variations imposed above. Without a capping inversion, there are wavenumbers for which these pressure perturbations are small, and the hydrostatic assumption might therefore no longer hold. In particular, we have noticed unphysical changes in ABL thickness for such cases.
   To clarify this in the manuscript, we changed line 122 to:

   "*Finally, we note that the hydrostatic assumption in the boundary layer ($\frac{\partial p}{\partial z} = 0$) is only reasonable as long as pressure effects within the ABL are negligible compared to those of the gravity waves. In particular in cases where a capping inversion is absent, we have noticed that this assumption may not be valid, which can lead to unphysical perturbations.*"

   (c) Line 130 How does it couple to an actuator disk model? Gaussain filter with L=1km
   The turbine forces are computed using an actuator disk model, with the wake-scale velocity gradients being incorporated using a wake model. These forces are then filtered onto the coarser TLM grid using a Gaussian filter with a length of 1km. To clarify this, we changed line 130 to:

   "*The turbine forces are computed individually using an actuator disk model.*"

   (d) Line 306 gives N=0.013 while the figure 3 caption gives N=0.0113. Are both correct?
   We thank the referee for noticing this mistake. The correct value is $N = 0.0113 s^{-1}$. Line 306 has been changed to reflect this.

   (e) Line 227 to 230 are unclear.
   We changed them to clarify any confusion:

   "*since evaluating $\Omega$ and its derivative evaluated at the center of each sublayer in equations equations 28 and 29, as the use of a constant wavenumber implies, results in a discontinuous profile for $W$. As Pütz et al. (2019) found, these discontinuities cause the piecewise method to converge to a different solution as solving equation 16 with a simple finite difference solver. However, the physical reasoning behind the kinematic and dynamic boundary conditions is generally valid, indicating that they should always be used.*"

   (f) The title could be shorter and more attractive
   We propose the following alternative:

   "*Including realistic upper atmospheres in a wind-farm gravity-wave model*"

---

## Author Response (AR2)

**1   Response to the editor**

- I want to congratulate you on satisfactorily carrying on all the revisions recommended by the two referees.
  We would like to thank the associate editor for the kind words, and for their help throughout the submission process.

- I want to suggest that you make the model code more accessible to possible researchers and generate a DOI by, for example, using GitHub and Zenodo. It is not a requirement of WES but, in my opinion, is a valuable asset for your research and the community.
  We agree with the suggestion, but we are currently not able to make the code open-source. However, we are working towards this, and an open-source version is planned to be released by the end of the year. We added a line in the *Code and data availability* section on line 486 to mention this.

- ..., subject to minor revisions, including revising all references; many are incomplete and lack DOIs. Also, the data source of the ERA5 data is missing in the manuscript; please check the data acknowledgement section of the dataset.
  We would like to thank the associate editor for noticing this. The reference list has been corrected, and a reference to the ERA5 data source has been added on line 428 and in the *Acknowledgements* section.